# The Dual Role of the Innate Immune System in the Effectiveness of mRNA Therapeutics

**DOI:** 10.3390/ijms241914820

**Published:** 2023-10-01

**Authors:** Albert Muslimov, Valeriy Tereshchenko, Daniil Shevyrev, Anna Rogova, Kirill Lepik, Vasiliy Reshetnikov, Roman Ivanov

**Affiliations:** 1Scientific Center for Translational Medicine, Sirius University of Science and Technology, Olympic Ave 1, 354340 Sirius, Russia; tereschenko.vp@talantiuspeh.ru (V.T.); dr.daniil25@mail.ru (D.S.); reshetnikov.vv@talantiuspeh.ru (V.R.); ivanov.ra@talantiuspeh.ru (R.I.); 2Laboratory of Nano- and Microencapsulation of Biologically Active Substances, Peter the Great St. Petersburg Polytechnic University, Polytechnicheskaya 29, 195251 St. Petersburg, Russia; anna.aroo@mail.ru; 3RM Gorbacheva Research Institute, Pavlov University, L’va Tolstogo 6-8, 197022 St. Petersburg, Russia; lepikkv@gmail.com; 4Saint-Petersburg Chemical-Pharmaceutical University, Professora Popova 14, 197376 St. Petersburg, Russia; 5School of Physics and Engineering, ITMO University, Lomonosova 9, 191002 St. Petersburg, Russia; 6Institute of Cytology and Genetics, Siberian Branch of Russian Academy of Sciences, Prospekt Akad. Lavrentyeva 10, 630090 Novosibirsk, Russia

**Keywords:** mRNA, innate immunity, NF-κB, PAMP, DAMP, cGAS/STING, NLRP3, adjuvants

## Abstract

Advances in molecular biology have revolutionized the use of messenger RNA (mRNA) as a therapeutic. The concept of nucleic acid therapy with mRNA originated in 1990 when Wolff et al. reported successful expression of proteins in target organs by direct injection of either plasmid DNA or mRNA. It took decades to bring the transfection efficiency of mRNA closer to that of DNA. The next few decades were dedicated to turning in vitro-transcribed (IVT) mRNA from a promising delivery tool for gene therapy into a full-blown therapeutic modality, which changed the biotech market rapidly. Hundreds of clinical trials are currently underway using mRNA for prophylaxis and therapy of infectious diseases and cancers, in regenerative medicine, and genome editing. The potential of IVT mRNA to induce an innate immune response favors its use for vaccination and immunotherapy. Nonetheless, in non-immunotherapy applications, the intrinsic immunostimulatory activity of mRNA directly hinders the desired therapeutic effect since it can seriously impair the target protein expression. Targeting the same innate immune factors can increase the effectiveness of mRNA therapeutics for some indications and decrease it for others, and vice versa. The review aims to present the innate immunity-related ‘barriers’ or ‘springboards’ that may affect the development of immunotherapies and non-immunotherapy applications of mRNA medicines.

## 1. Introduction

With the broad clinical application of mRNA-based therapeutics, mRNA technology has made a real breakthrough in medical biotechnology, regenerative medicine, replacement therapy, and genome editing [1,2]. Although the principle of using in vitro-transcribed (IVT) mRNA in gene therapy was described many years ago, its actual clinical application became possible only in recent years, as non-viral delivery of nucleic acids was extensively developed [2,3]. To date, mRNA is regarded largely as a promising tool for preventive and therapeutic immunization.

The fundamental mechanism underlying the mRNA vaccine technology is based on a vector that delivers a nucleic acid molecule encoding a gene of interest into a target cell, thus allowing the cell to produce the target protein and to express the antigen to elicit an immune response [4]. mRNA vaccines have potential benefits over other nucleic-acid-based therapies, such as viral vectors or plasmid DNA. mRNA does not need to penetrate nucleus barriers and thus it does not lead to irreversible changes to the cell or its genome. It provides almost immediate initiation of protein translation as soon as it appears in the cytoplasm. In contrast with viral vaccines, mRNA-based therapeutics do not lead to the production of infectious particles and have no potential to cause adverse effects such as immune-mediated hepatitis [1,5,6].

When applying mRNA technology, one should consider molecular and biological consequences of the cellular response to foreign RNA. If a eukaryotic cell detects foreign RNA, it responds to the viral invasion. The innate immune system has nucleic-acid sensors, RNA sensors in particular, which constantly scan for and detect foreign RNA and endogenous RNA of cells under stress. Numerous RNA-binding proteins in mammalian cells control immune reactions by modulating gene expression, splicing, nuclear export, mRNA modification, translation, and degradation [7].

Multiple molecular mechanisms evolved to distinguish self and non-self RNAs to secure the initiation of an antiviral response, including the expression of type I interferon (IFN) and IFN-stimulated genes. The main RNA response mechanism is related to the recognition of pathogen-associated molecular patterns (PAMPs) by pattern recognition receptors (PRRs). PRRs can be categorized into several families based on structure, localization, and ligands [8]. They include toll-like receptors (TLRs), C-type lectin receptors (CLRs), retinoic acid-inducible gene I (RIG-I), melanoma differentiation-associated protein 5 (MDA5), the cyclic GMP-AMP synthase (cGAS)/stimulator of interferon genes (STING) second messenger system, and NOD-like receptors (NLRs). Usually, mRNA enters the cell by endocytosis. There are specific receptors in endosomes that can recognize RNA, such as TLR3 for double-stranded RNA (dsRNA) and TLR7/8 for single-stranded RNA (ssRNA). Other receptors in the cytosol—e.g., RIG-I, MDA5, NLRP3, and NOD2—recognize exogenic RNA [9].

The recognition of RNA activates transcription factors called IRF3 and NF-κB. NF-κB induces inflammation, whereas IRF3 inhibits the translation of exogenous RNA via interferon synthesis [9,10]. IFNs regulate various cell functions by inducing expression of IFN-stimulated genes, e.g., signal transduction pathways of Janus kinases and transcription activators [10]. The nuclear factor NF-κB plays a major role in a wide range of cell functions, including launching the innate immune response mechanism that targets viral nucleic acids [10]. NF-κB is found in various mammalian cell types, including immune cells such as macrophages and dendritic cells.

NF-κB activation directly impacts the function of immune cells [11], such as the differentiation of dendritic cells into antigen-presenting cells, T-cell differentiation, recruitment of neutrophils to an inflammation site, and macrophage polarization. At the supramolecular level, NF-κB plays a pivotal part as a transcription factor. When activated, it induces transcription of pro-inflammatory genes [11,12], leading to the synthesis of cytokines, chemokines, cell cycle regulators, anti-apoptotic factors, and adhesion molecules [10,11,12] (see Table 1).

The ability of RNA to induce an immune response through activation can be regarded both as an advantage and a limitation of RNA-based therapeutics. On the one hand, an mRNA vaccine is an adjuvant itself that facilitates the survival and proliferation of immune cells presenting the target antigen. On the other hand, the inappropriate activation of the innate immune response (mostly due to IFN) can lead to insufficient expression of the target gene and weaken the mRNA’s effect in non-immunotherapy settings [20]. In the text below, we will focus in detail on the specific pathways of innate immune response activation in the context of the development and clinical application of mRNA-based therapeutics (Figure 1).

## 2. Innate Immune Response Pathways Recognizing mRNA

### 2.1. RIG-I and MDA5 Signaling

PRRs such as helicase proteins RIG-I (retinoic acid-inducible gene I) and MDA5 (melanoma differentiation-associated protein 5) are key participants in the early phase of the immune response to viral infections. RIG-I and MDA5 (together called RIG-I-like receptors or RLRs) are localized to the cytoplasm in the phosphorylated state and inactive conformation [21,22]. Binding to PAMP (usually with dsRNA or other types of RNA [22]) sequentially leads to conformational transformation, dephosphorylation of the C-terminal domain (CTD) and the N-terminal caspase activation and recruitment domains (CARD1 or 2), and ubiquitination [23,24]. The activation of RIG-I and MDA5 is accompanied by their oligomerization on dsRNA, with ubiquitin marks playing a pivotal role in this process [25,26].

Next, the oligomerized and ubiquitinated complexes of RIG-I or MDA5 with dsRNA interact through CARD domains with mitochondrial antiviral signaling (MAVS) proteins [27], which consequently activate transcription factors NF-κB and IFN-regulatory factors IRF3/7 through the assembly of kinase complexes IKKα/β/γ and TBK1/IKKΣ (TANK-binding kinase 11/inhibitor of nuclear factor kappa-B kinase subunit epsilon), respectively [28,29]. NF-κB triggers the expression of IL-1β and IL-6, IRF3 triggers the expression of IFNβ and INFγ, while IRF7 initiates the expression of IFNα [30]. IFNs are secreted out of the cell and bind to their receptors, resulting in a positive feedback signaling loop and increasing local secretion of IFNs and other immunostimulatory factors, leading to the antiviral response, RNA degradation, and suppressed translation [31].

Laboratory of genetics and physiology 2 (LGP2) is the third and least well-understood member of RLRs. It modulates the function of RIG-I and MDA5 during viral infection in two opposite ways. On the one hand, it works with MDA5 to increase the sensing of RNA viruses. LGP2 primarily binds the ends of dsRNA or even coats the molecule and then incorporates itself into MDA5 filaments. Thus, it enhances the interaction between MDA5 and RNA, which leads to an increase in downstream signaling and a type I IFN response [32,33]. On the other hand, LGP2 suppresses RIG-I signaling by applying different mechanisms, such as immunostimulatory RNA sequestration, prevention of RIG-I binding to MAVS, and inhibition of RIG-I ubiquitination [32,34].

While developing mRNA vaccines, one has to consider the activation of RIG-I and MDA5 because their main ligand—dsRNA—is a typical side product of IVT. In the past few years, IVT has been used as a dominant technology in mRNA production for therapeutic applications. The phage RNA polymerase, the pivotal enzyme used in IVT, is known to have RNA-dependent RNA polymerase activity and takes part in the assembly of ssRNA higher-order structures [31]. Additionally, phage RNA polymerase generates 5′-phosphorylated RNA, which is also typical for viruses and whose double-stranded version is a powerful RIG-I agonist [22,35].

Recent studies showed that modulating the activity of RLRs and other PRRs can significantly boost the yield of the target protein, ensuring a better efficacy of both non-immunotherapeutic modalities. For example, the reduced activation of RIG-I and other PRRs due to the chemical modification of mRNA leads to a 4-fold rise in erythropoietin expression and a significant increase of hematocrit and prevents the production of surfactant protein B (SP-B), which is lethal for mice [36].

The stimulation of RIG-I 5′ppp-RNA enhances the protective antiviral response against influenza (IAV-influenza virus) and chikungunya virus [37]. The stimulation of MDA5 leads to a 40% tumor growth delay in mouse models [38]. The available data about the mechanisms of RIG-I and MDA5 action and about their similarities and differences open up the opportunity for regulating their activity in order to improve the efficiency and safety of non-immunotherapeutic applications as well as vaccines against infectious diseases and cancer.

As a starting point for the RIG-I and MDA5 activity control strategy, one can employ information about the size of the therapeutic mRNA in question. It has been established that dsRNAs activate RIG-I and MDA5 in different ways: RIG-I preferentially recognizes short sequences (no longer than 2 kb), and MDA5 recognizes long dsRNA sequences [39]. 

At the same time, RIG-I initiates IFN synthesis in response to at least 20 bp of dsRNA in length [40], while MDA5 requires at least 100 bp [41]. For the RIG-I stimulation, one can use miRNA and siRNA that are 20–24 bp in length. Hence, the signaling pathway in question can be determined based on the size of the RNA to be delivered.

Another factor in controlling RLR activation is the desired duration of the therapeutic effect from delivered mRNA. In the case of vaccines against communicable diseases, where mRNA plays the role of adjuvant and thus should not stay in the cell for long, modulation of the RIG-I is required. It is RIG-I that is responsible for the initial immune response to exogenous RNA [42]. In contrast, the stimulation of MDA5 could come in handy when we need a prolonged therapeutic effect, as in the production of self-amplifying RNA for non-immunotherapeutic applications. MDA5 maintains anti-inflammatory signaling for up to 5 days [43].

A precise understanding of RIG-I and MDA5 signaling pathways can be useful for designing therapeutic vaccines against specific communicable diseases. Consequently, NF-κB activation via an IκBα-independent pathway, which is typical for MDA5-signaling [22], can be used to overcome the NF-κB pathway suppression, related to a decline of IκBα phosphorylation associated with *Mycobacterium tuberculosis* infection [44]. The activation of RIG-I in turn initiates caspase-1–dependent inflammasome activation and processing of mature IL-1^®^ [45], which can also compensate for some processes, as observed in tuberculosis [46].

The activation of RIG-I and MDA5 in non-immunotherapeutic approaches can be diminished with standard methods for decreasing the immunogenicity of RNA, in particular by capping, reducing the uridine (U) content via nucleoside modifications, polyadenylation, reducing the number of double-stranded products of in vitro transcription, and by inhibition of PRRs [47]. As a specific way to decrease RLR signaling, one can use the cap-1 structure, which prevents RNA from binding to both RIG-I and MDA5 [48,49]. Another possible way to reduce the activity or even inhibit RIG-I completely is to introduce at least a single unpaired 5′ triphosphorylated nucleotide or a 3′ overhang (next to a 5′ triphosphate end at a dsRNA terminus) and to reduce the number of phosphate groups at the 5′ end [22].

Viral proteins, for example, NS-1 influenza protein and protein V in paramyxoviruses, are capable of inhibiting RIG-I and MDA5, respectively [50,51], and such viral proteins can reduce immunogenicity if delivered together with RNA. Viral RNA, its derivatives, and imitation of viral RNA structure can be regarded as possible stimulators of RLR activity and could be used as antivirals, cancer therapeutics, and vaccine adjuvants [52]. It has been demonstrated that an optimal protective effect can be achieved with the simultaneous activation of RIG-I and MDA5 [53,54].

At the same time, the immune response can be modulated. For instance, the administration of polyinosinic:polycytidylic acid [poly(I:C)] results in a response with the prevalence of IFN III [55] and can help to minimize the adverse effects of IFN I activation [56].

### 2.2. TLR Signaling

Toll-like receptors (TLRs) are a major group of receptors that belong to the PRR family [57,58,59]. TLRs are the transmembrane proteins capable of recognizing molecules frequently found in pathogens (PAMPs) or molecules released by damaged cells (damage-associated molecular patterns: DAMPs) [59]. TLRs can be engaged by various viral, bacterial, and mucosal structures as well as components of damaged or dying cells [58,59,60,61,62]. They trigger the signaling cascades that amplify the transcription of immune-defense genes [62,63]. Considering the constitutive membrane expression on various cell structures, TLRs can be viewed as innate-immunity guards against invading microbial pathogens at the frontier of host defense [64].

TLRs consist of two domains: an extracellular transmembrane domain and an intracellular C-terminal domain (Figure 2) [65]. The N-terminal domain, also known as an ectodomain, is located outside the plasma membrane and has the shape of a horseshoe. It acts as a receptor and contains 19–25 highly conserved leucine-rich repeats (LRRs), each consisting of 24–29 amino acid residues [66,67]. A similar structure can be found in variable receptors of lymphocytes in early vertebrates [68]. Tandem LRRs have also been discovered in certain bacteria [69] and most plants, where they are a part of the protein structures responsible for infection resistance [70,71]. This observation highlights evolutionary conservatism and structural universality in terms of recognition of PAMPs and DAMPs. Notably, glycosylation of the TLR ectodomain has an impact on ligand–receptor interaction [72].

The intracellular C-terminal domain consists of ~150 amino acid residues and is called Toll/interleukin 1 receptor (IL-1R) homology or TIR. The domain recruits adapter proteins and launches intracellular signals [73,74]. To date, 10 human subtypes of TLRs have been identified [75]. Although the functions of different TLRs may overlap, each is characterized by a specific set of recognized PAMP or DAMP molecules. Some TLRs, such as TLR1, TLR6, and TLR10, can form homo- or heterodimers, but the biological role of these complexes is yet to be elucidated [76].

TLRs are located on the external cell membrane and the surface of intracellular vesicles, endosomes, lysosomes, endolysosomes, and other structures. On the surface of the cell, there are TLR1, TLR2, TLR4, TLR5, TLR6, and TLR10, and on the inside of the cell, there are TLR3, TLR7, TLR8, TLR9, TLR11, TLR12, and TLR13 [77]. TLR2 and TLR4 can also be found in dendritic and epithelial cells [78]. The location of TLRs is determined by their functions and depends on the types of pathogens recognized.

The binding of TLRs to a ligand leads to the internalization of the complex and proteolytic cleavage of the receptor ectodomain; this process is required to initiate TLR signaling [62,79]. The engagement of TLRs initiates a complicated cascade of biochemical reactions that starts with the C-terminal TIR domain [80]. This domain interacts with similar TIR domains of adaptor proteins, which determine the specificity of TLR signaling by their type [77,81].

A collection of PAMPs, which characterizes a given pathogen, stimulates a specific set of TLRs. In this way, the resultant PAMP signal affects the speed and intensity of the immune response and determines the balance among major populations of T helper cells (Th1/Th2/Th17/Treg) [82]. In other words, due to the small diversity of TLRs, the innate immune system is capable of recognizing different groups of pathogenic and symbiotic microorganisms. In most cases, this arrangement enables the immune system to choose the optimal type of secondary immune response required for homeostasis [83,84].

The activation of interferon-regulatory factors (IRFs) and NF-κB are the key steps of the TLRs’ signaling pathway [85]. The activation of NF-κB induces the expression of proinflammatory cytokines, while the activation of IRFs provides optimal production of type I IFNs [86].

The production of IFNs works simultaneously in two directions by hindering and facilitating mRNA vaccines. On the one hand, it induces activation of protein kinase R (PKR) and 2′-5′-oligoadenylate synthetase (OAS), and these systems inhibit translation and cause degradation of cellular and ribosomal RNA, respectively, consequently decreasing the efficiency of mRNA-based products [87,88]. On the other hand, the elevated production of IFNs stimulates the immune response against the target antigen and this effect may be useful in the case of mRNA vaccines [89].

Discussing the development of mRNA-based therapeutics, TLRs are of particular interest because they are located on different components of cells and are responsible for different ways of antiviral immunity. TLRs can significantly affect the efficacy of mRNA-containing products [90,91]. Except for TLR1 and TLR5 (which recognize bacterial lipoproteins and flagellin), almost all subtypes of TLRs are involved in the recognition of viral particles [92,93]. Namely, TLR2, TLR4, TLR6, and TLR10 recognize glycoproteins of the virus envelope and some viral enzymes. At the same time, TLR3 recognizes dsRNA, whereas TLR7 and TLR8 recognize ssRNA, and TLR9 recognizes ssDNA [93]. Synthetized mRNA preparations contain dsRNA impurities in addition to ssRNA and thus can be recognized by TLR3, causing sustained production of type I IFN [94]. Exogenous ssRNA, entering the cell within a therapeutic formulation, can also cause the production of IFN-I by engaging TLR7 and TLR8 [95,96].

During evolution, many viruses have developed ways to resist the host immune surveillance mechanisms [97]. Such mechanisms include inhibition of TLR synthesis in the host’s cells [98], a reduction in TLRs’ activity through disruption of their plasma membrane localization [99], and various ways to impair adapter proteins of TLRs [100,101]. Some viruses have developed proteins that, by binding to TLRs, can suppress their function directly, reducing IFN production and disrupting the NF-κB pathway [101,102,103]. One can use these mechanisms to create therapeutic mRNA products, which control the amplitude of an immune response or shift the balance of T helper populations (Th1/Th2/Th17/Treg). Thus, TLR agonists are well tolerated and significantly increase the efficacy of mRNA vaccines.

Moreover, the FDA has recently approved TLR2, TLR4, and TLR9 agonists as adjuvants for the BCG vaccine [104,105,106]. The latest SARS-CoV-2 pandemic has spurred the development and use of mRNA vaccines that have shown high safety and efficacy [107]. Such promising results are based on almost four decades of research, having led to the development of reliable methods for protecting artificial mRNA from degradation and enhancing its translation in the host cell [108,109,110]. Early methods included 5′-capping, 3′-polyadenylation, various base modifications such as 1-methyl pseudouridine (m1Ψ), and the use of an optimized Kozak consensus sequence in the 5′UTR region, etc. [111,112,113,114].

Over the last two decades, a number of agonists have been designed for almost all types of TLRs, and some of them have reached clinical use as vaccine adjuvants [115,116,117]. This, together with emerging methods for protecting mRNA against degradation, opens bright prospects for modulation of the primary immune response through selective action on various members of the TLR family [118] as well as extensive possibilities regarding manipulation of various components of innate immunity.

Artificial mRNA is characterized by its own adjuvant activity because it can stimulate TLR7/8. This effect can be enhanced by additional agonists, such as protamine, a cationic TLR7 activator protein that will enhance the production of type I IFNs and potentially increase the amplitude of a secondary immune response [118].

Karikó et al. showed that removing dsRNA from the mRNA mixture using high-performance liquid chromatography enhances translation of the target mRNA up to 1000-fold. Moreover, the purified mRNA causes less induction of IFNs and inflammatory cytokines [94]. This study unveils two significant points. Firstly, chromatography application in the production of mRNA vaccines helps to improve significantly the quality of final vaccine preparation. The technique aims to purify samples from non-target nucleic acids that are, among other things, ligands for TLRs. In light of such advanced purification methods, there are ongoing discussions regarding whether base modifications are necessary, as the level of translation enhancement is the greatest when mRNA is unmodified. The second point refers to the role of dsRNA in an immune response. Compared with other types of exogenous RNAs, it is the main trigger of cellular response. Hence, one should keep in mind both points when creating mRNA vaccines.

Alternatively, a reduction of immunogenicity can be attained by using modified bases such as pseudouridine or m1Ψ, which change mRNA secondary structures (hairpins) and thus prevent receptor TLR3 or RIG-I from recognizing mRNA [119]. As a result, TLR3 continues downregulating the production of type I IFNs [120,121]. Of note, this approach shows better performance than chromatographic purification for the removal of dsRNA from the original product. Because of the large amount of RNA in the cell, RNA polymerase can synthesize a small number of antisense oligonucleotides and trigger TLR3. Thus, the incorporation of modified bases into mRNA decreases to a certain extent its immunogenicity, avoiding its degradation and eliminating the need for its thorough purification [94,122,123,124].

Furthermore, modified RNA bases can change mRNA’s secondary structure and, therefore, prevent its recognition by TLR3 and RIG-I receptors [119]. There is also indirect evidence that the presence of m1Ψ in the mRNA structure alters the steric organization of hydrogen bonds and prevents them from binding to TLR7/8 [119,123]. In this way, the incorporation of the m1Ψ nucleoside allows, in a sense, to hide RNA from immune surveillance and extend its lifetime in the cell, thereby potentially increasing the protein yield.

Nevertheless, it would be naive to believe that the steric changes associated with the introduction of m1Ψ will not alter the dynamics of the translation process [125]. Although most research has documented a significant enhancement of protein expression [126], free energy calculations have shown that the presence of m1Ψ in the mRNA strand may negatively affect protein synthesis by reducing the thermodynamic efficiency of interaction with the ribosome [127,128,129]. This assumption has later been experimentally confirmed only for some artificial mRNAs [130]. Thus, the incorporation of m1Ψ into the mRNA strand allows it to extend its lifetime significantly and prevents the activation of TLRs and RIG-I, thereby significantly reducing immunogenicity.

The development of various delivery systems opens up opportunities for the simultaneous use of several TLR agonists and non-TLR adjuvants in one mRNA product [109,115,131,132]. This combined approach helps to shape a desired immune response [133,134]. For example, simultaneous stimulation of TLR7/8 and TLR/9 can elicit a strong Th1 response and a high level of IgG2a antibodies [133,134]. In the meantime, mRNA administration, together with simultaneous stimulation of TLR4 and TLR9, elicits a strong Th2 response and a high level of IgG1 [133,134]. Hence, this evidence confirms the possibility of using combinations of TLR agonists with various mRNA constructs in one therapeutic product to induce an immune response with specified parameters.

### 2.3. NOD-like Receptors

Another equally important group of PRRs is NOD-like receptors (NLRs). This is a conserved family of intracellular receptors that are thought to be sensors capable of recognizing DAMPs and PAMPs in the cytoplasm [135]. To date, 22 NLR genes have been identified in humans [136].

The proteins of the NLR family share many structural features, whereas the structure, in general, is similar to that of TLRs [137,138]. All NLRs consist of three domains, two of which (C-terminal and central domains) are common for all members of the family [8] (Figure 3). The extracellular C-terminal leucine-rich repeat (LRR) domain has the form of a horseshoe. The central nucleoside-binding domain is a component of the larger NACHT domain. The NACHT domain oligomerizes during NLR activation using ATP energy and interacts with an N-terminal effector domain.

The N-terminal domain differs across NLRs and divides the family into subfamilies [135,136,139,140]. There are four different subfamilies: NLRA, NLRBs (also known as NAIPs), NLRCs, and NLRPs. The NLRA subfamily contains an acidic transactivating domain. CIITA (MHC class II transcription activator) is the only member of the NLRA subfamily. The NLRB subfamily contains an N-terminal BIR-like domain. NAIP is the only protein of the subfamily detected in humans.

The two most prominent of these subfamilies are NLRC and NLRP. The NLRC proteins, namely NOD1, NLRX1, NOD2, and NLRC3–5, have one or more caspase activation and recruitment domains (CARD). NLRP proteins have pyrin domains and are the best characterized among other subfamilies. This subfamily consists of 14 members named NLRP1–14 [8,135,136].

NLRs are broadly categorized into two groups according to their ability to form inflammasomes. The first group is called canonical and signals through the formation of caspase-1–activation inflammasomes. Another group is called non-canonical because NLR proteins act through either an alternate caspase or an inflammasome-independent mechanism. Of particular interest are NLRPs, whose function is well-studied [8]. Some proteins, such as NLRP3, have both canonical and non-canonical functions.

NLRs take part in antigen presentation (NLRC5 and CIITA), pathogen/damage sensing (NOD1-2, NLRC4, NAIP, and NLRP3), inflammasome activation (NLRP3, NLRP1, NLRP2, NLRP6, and NLRP7), and inflammatory signaling inhibition (NLRC3, NLRC5, NLRX1, and NLRP6) [141]. Furthermore, Lupfer and Kanneganti showed that NLRP2, NLRP5, and NLRP7 are involved in non-immune pathways such as embryonic development [142]. The study highlighted alternative functions for some NLRs in different cell types and a multitude of activation mechanisms that provide independent downstream effects for other NLRs.

In the next section, we are going to discuss inflammasome activation in detail on the example of NLRP3 inflammasome activation. Here, it is worth discussing other NLR functions that could be relevant to the development of mRNA vaccines.

The CIITA expression is activated by the expression of IFNγ and is regulated mostly at a transcriptional level [143]. It acts as a transcriptional coactivator implicated in the regulation of MHC class II expression. However, different viruses and other pathogens, such as HIV, VZV, human cytomegalovirus, Epstein–Barr virus, and mycobacteria, can inhibit CIIPA and thereby suppress MHC II gene expression [143,144,145,146,147,148].

The NLRC5 expression is also regulated by IFNγ and can be additionally induced (although less effectively) by type I IFNs, LPS, polyinosinic:polycytidylic acid [poly(I:C)], and some viruses [149,150]. It was first identified as a transcriptional regulator of MHC class I genes and referred to as CITA or class I transactivator [151]. At the same time, there was quite contradictory data showing that NLRC5 may play both negative and positive roles in type I IFN production [150]. On the one hand, NLRC5 was shown to interact directly with NF-κB or RIG-I and repress TLR4 signaling and type-I IFN responses. Meanwhile, it could enhance type I IFN production [152].

NOD2 is known to control cytokine expression through its activation of NF-κB and p38 MAPK-dependent signaling in response to bacterial infection. However, it also takes part in the response to exogenous dsRNA and ssRNA [153]. dsRNA molecules induce OAS-1 to interact with NOD2. In turn, this leads to the production of 3′-5′ oligoadenylate synthetase type 2, which activates antiviral responses, particularly RNase L activation.

Furthermore, it was shown that NOD2 functions as a cytoplasmic viral PRR and recognizes viral ssRNA. It utilizes the adaptor protein MAVS to activate IRF3, leading to the production of IFNβ [154]. Other NLRs, such as NOD1, NLRC4, NAIP, or NLRC3, did not show the ability to activate IRF3 in response to ssRNA. Hence, NOD2 utilizes either RICK or MAVS depending on the stimulus (e.g., MDP or. ssRNA) to activate either NF-κB and MARK or IRF3, respectively. It is worth mentioning that NOD1 also activates NF-κB and MAPK signaling pathways, recognizing bacterial γ-D-glutamyl-meso-diaminopimelic acid [137,155]. NLRX1 is a negative regulator of MAVS signaling during viral dsRNA and ssRNA invasion [153,156]. This NLR protein is interesting as it is located on the mitochondria and contains mitochondrial targeting sequences. NLRX1 interacts with MAVS and prevents its binding to RIG-I. This interaction leads to disruption of the activation of NF-κB and IRF3 and the inhibition of type I IFN and proinflammatory cytokine production [153].

One could consider modified muramyl dipeptide molecules as potential adjuvants for mRNA vaccines [157]. These molecules are produced during bacterial peptidoglycans degradation, and they induce NALP3-mediated activation of caspase-1 and maturation of proIL-1β [158]. The muramyl dipeptide molecules also require NOD2 to boost humoral and cellular immune responses to their optimum level.

Several studies show that the cooperation of NLRs and TLRs is worth considering when developing mRNA vaccines. NLR stimulation in various combinations with TLRs causes stable polarization of an immune response and shifts the balance between Th1/Th2 populations. The synthetic peptide FK-156 alone can stimulate NOD1, launching a Th2 response [159,160]. Furthermore, the bi-functional ligand CL-429 can activate both NOD2 and TLR2 and, therefore, suppress virus-induced inflammation [161].

NLR agonists lead to persistent changes in the innate immune system affecting immunity to various infections [157,162]. Such “memory” of innate immunity is called trained immunity [163,164], in which PRRs play a substantial part. It was shown that bacillus Calmette-Guérin (BCG) interacts with TLR2 or TLR4 and signals via NOD2 pathways, leading to the production of IL-1β, TNF, IL-6, and IL-32 cytokines [165]. The combined effect on various types of PRRs in the context of the application of mRNA opens vast opportunities for managing immune responses.

### 2.4. NLRP3 Inflammasome Activation

As a representative of the nucleotide-binding leucine-rich repeat proteins (NLRPs) family, the NLRP3 protein mainly contains three components: an LRR domain, the adenosine triphosphatase (ATPase) domain known as NACHT, and the N-terminal pyrin domain (PYD) [166,167].

NLRP3 responds to numerous stimuli, including viral/bacterial DNA and RNA, pore-forming toxins, silica, asbestos, uric acid, ion flux, and ATP [168]. Owing to such a broad range of NLRP3 activators, direct binding of all these activators to NLRP3 seems unlikely. Recruitment of different cofactors might help to determine ligand specificity.

Once the LRR domain detects a pathogen or an endogenous danger signal, the oligomeric NLRP3 inflammasome binds its subunits through NACHT domains. Then, the complex recruits ASC through PYD–PYD interactions thereby nucleating PYD filaments of ASC. The adaptor protein ASC attracts pro-caspase 1 via CARD–CARD interactions and induces the self-cleavage of caspase 1 [169].

The activation of the NLRP3 inflammasome has been described as a two-step signal model. The first signal (priming) is provided by TLR and cytokine receptors, such as the interleukin 1 receptor (IL-1R) and tumor necrosis factor receptor (TNFR) [170]. After the recognition of microbes or inflammatory cytokines by NF-κB-activating receptors, NF-κB is immediately translocated to the nucleus with the assistance of FADD and caspase 8. They raise the amount of NLRP3 and pro-IL-1β by boosting transcription of relevant genes and translation of the resultant mRNAs [170,171].

The second signal (activation) is triggered by diverse stimuli, including pore-forming toxins, ATP, and different particulates [172,173]. Owing to the second step, the NLRP3 inflammasome completes the assembly and activates caspase 1. The latter processes pro-IL-1β and pro-IL-18 into mature proteins at the microtubule-organizing center, distributed in the perinuclear and punctate regions [166,174].

It has been suggested that NLRP3 inflammasome activation is mediated by intricate cellular signaling events, including potassium efflux, calcium overload, reactive oxygen species (ROS) production, mitochondrial dysfunction, and lysosomal rupture [175,176,177,178,179]. Yet, the specific mechanism underlying ion flux changes or organelle dysfunction during the activation of the NLRP3 inflammasome remains controversial.

Available experimental data suggests that human macrophages sense all bacterial RNA components (mRNA, tRNA, and rRNAs) and synthetic ssRNA, activating the NLRP3 inflammasome, whereas murine cells preferentially recognize bacterial mRNA [180]. Other studies have shown that DHX33 launches the assembly of NLRP3–ASC–caspase 1 by binding directly to polyI:C dsRNA or to *E. coli* total RNA [181].

A number of recent articles have linked NLRP3 inflammasome activation to mitochondrial dysfunction. In particular, the production of reactive oxygen species by phagocytosis or mitochondria is thought to be critical for the activation of the NLRP3 inflammasome in response to various stimuli, including microbial RNAs and endogenous noncoding RNA transcripts of the Alu retrotransposon [168,182].

The dysregulated NLRP3 inflammasome has been implicated in the pathogenesis of several autoimmune and autoinflammatory conditions, infections, and tumorigenesis [183,184]. Elucidating the role of the NLRP3 inflammasome in the pathogenesis of various disorders is a matter of ongoing research. Undoubtedly, a further clarified and precise understanding of the molecular mechanism behind the NLRP3 inflammasome regulation will guide the development of new effective therapeutics.

Being a key component of inflammatory response and pyroptosis pathways, NLRP3 activation has a detrimental effect on mRNA transfection efficiency [185]. According to clinical data on the administration of the mRNA-1273 COVID-19 vaccine [186], this activation also may contribute to some adverse events of mRNA therapeutics.

As activated TLRs prime an NLRP3-mediated response, potential contaminants, such as LPS and bacterial RNA components (mRNA, tRNA, and rRNAs), should be removed. Furthermore, the formulation of mRNA vaccines should be carefully examined to avoid NLRP3 activation.

Forster Iii J et al. have tested a panel of six mRNA lipid nanoparticle (LNP) formulations while varying concentrations of different lipid components. Of particular interest were nanoparticles containing a high concentration of the ionizable DLin-MC3-DMA lipid in tandem, highly cationic lipid DOTAP, and a low cholesterol concentration. Such nanoparticles dramatically activate the NLRP3 inflammasome, with associated cytotoxicity and decreased GFP transfection efficiency [185]. These data indicate that optimization of key lipid concentrations in mRNA LNPs partially affects the degree of NLRP3 inflammasome activation. Hence, one should examine the lipid composition of an mRNA transfection tool in order to balance endosomal escape, inflammasome activation for disease-specific expression, and the development of self-adjuvant mRNA delivery systems.

Owing to the extensive participation of the NLRP3 pathway in several inflammatory diseases and cancers, a variety of chemical NLRP3 inhibitors has been proposed, with some of them tested in a clinical setting [173]. Nevertheless, because the usefulness of this pathway for the field of mRNA therapeutics only recently received well-deserved attention, our literature search did not reveal comprehensive data on the influence of known NLRP3 inhibitors (including Glyburide, 16673-34-0, FC11A-2, VX-765, VX-740, BHB, MCC950, CY-09, tranilast, OLT1177, oridonin, parthenolide, and Furanochalcone Velutone) on mRNA transfection efficiency [187]. A notable exception is BAY 11-7082, which results in a reproducible increase in the expression of OCT4 upon synthetic mRNA transfection into human skin cells [188]. Therefore, the utility of NLRP3 inhibitors for mRNA gene therapies is a promising subject for further research.

The central role of NLRP3 in inflammatory responses to various external stimuli has made this complex a common target for the development of vaccine adjuvants. Among clinically relevant NLRP3 adjuvants, there are aluminum salts [189], saponins [190], and emulsions that commonly include water-in-oil (W/O) and oil-in-water (O/W) such as prototypes Montanide 720 and MF59 [164], TLR activators such as TLR4 agonists MPL-A or GLA [191], TLR3 ligands [dsRNA analogs such as poly(I:C)], TLR5 agonists (flagellin), TLR7/TLR8 agonists (imidazoquinolines such as imiquimod), TLR9 activators (CpG oligodeoxynucleotides), chitosan [192], trehalose-6,6′-dibehenate [193], calcium phosphate nanoparticles [194], and PLGA [195].

To sum up, the importance of NLRP3 as the target of conventional vaccine adjuvants is generally recognized, and this complex is studied extensively. Nonetheless, its activation by mRNA therapeutics should be wisely balanced via tuning of transgene structure and carrier composition.

### 2.5. C-Type Lectin Receptors

C-type lectin receptors (CLRs) are a superfamily of proteins, which, in the broader sense, contain a C-type lectin-like domain [196]. These receptors can be divided into two groups based on their ability to recognize carbohydrate or non-carbohydrate ligands. Those that bind carbohydrates function in either a Ca^2+^ dependent or independent manner. CLRs are transmembrane and soluble pattern recognition receptors. They are expressed mostly by myeloid cells, such as dendritic cells (DC) and macrophages, and recognize both PAMPs and DAMPs [197,198]. Binding CLRs to PAMPs induces signaling, internalization of a pathogenic organism, its lysosomal degradation, and presentation of the epitopes to lymphocytes.

We are going to discuss a particular group of CLRs, capable of recognizing specific carbohydrate sequences. In terms of therapeutic mRNAs, special attention should be given to dendritic cells immunoreceptor (DCIR), blood dendritic cells antigen 2, CD303 (BDCA-2), and dendritic cell-specific ICAM-3 grabbing non-integrin (DC-SIGN) (Figure 4).

Apart from carbohydrate patterns of a pathogen, DCIR and BDCA-1 are capable of binding ssDNA [199]. Signaling from both receptors leads to the inhibition of the inflammatory response. DCIR, containing the immunoreceptor tyrosine-based inhibitory motif (ITIM), inhibits TLR8-mediated production of IL-12 and TNF through effects on various signaling pathways [200], CpG-ODN-induced expression of IL-1^®^ and IL-6 [201], and STAT1-dependent production of IFN I under the influence of *M. tuberculosis* during the infection [202].

BDCA-2 transmits the signal through the immunoreceptor tyrosine-based activation motif (ITAM) of the adaptor molecule FcRy [203]. In contrast to most cases, this process launches the anti-inflammatory signals that perhaps are related to the activation of phospholipase Cγ2 (PLCy2) and consequent mobilization of Ca^2+^, which inhibits activation of a protein called myeloid differentiation primary response 88 or MyD88, responsible for TLR-mediated production of cytokines [204]. BDCA-2 ligation is associated with the suppression of IFN I production and other proinflammatory cytokines that are triggered by TLR7 and TLR9 signaling [203,205]. In this way, the commonality of recognizable patterns of pathogenicity (ssRNA) and the functional antagonism of known receptors of the CLR and TLR families enable investigators to fine-tune an immune response when administering therapeutic RNA [199,204].

In the manufacturing of vaccines against infectious diseases and cancer, it is worth considering the structure of the therapeutic RNAs because this biomolecule should engage TLR7/8 and 9 rather than suppress DCIR and BDCA-2. For non-immunological applications, the increased stimulation of DCIR and BDCA-2 helps to overcome activation signals from TLR and thus to increase the therapeutic effect. Unfortunately, little is known about the specific RNA that would allow for separate stimulation of TLR and CLR. It is a matter of future research.

When producing vaccines against specific infectious diseases, one should keep in mind that CLRs contribute to the polarization of an immune response because of ongoing modulation of the cytokine response. For instance, eliminating DCIR activity restores the production of IL-12 and shifts the immune response toward Th1, which favorably affects the course of M. tuberculosis infection [202].

Strikingly, it was shown that CLRs such as DC-SIGN are involved in virus transmission [202]. For instance, DC-SIGN can mediate the release of HIV-1 virions associated with exosomes and help their dissemination to CD4+ T cells. While binding to DC-SIGN, the HCV particles are kept from lysosomal degradation in immature DC and transmitted to hepatocytes [206].

### 2.6. The cGAS/STING Pathway

The cyclic GMP–AMP synthase (cGAS)/stimulator of IFN genes (STING) pathway emerged as a critical mechanism linking DNA sensing to induce robust innate immune defense programs [207]. Its activation lacks any pathogen-specific attributes, which sets it apart from several other innate immune signaling mechanisms [208]. Within this pathway, the binding of cGAS to double-stranded DNA (dsDNA) allosterically activates its catalytic activity. Activated cGAS leads to the production of 2′-3′-cyclic GMP–AMP (cGAMP), a secondary messenger molecule and potent agonist of STING [209]. Aside from cGAMP, STING also directly senses other cyclic dinucleotides secreted by some bacteria [210].

cGAMP binds to a stimulator of IFN genes (STING) dimers localized at the endoplasmic reticulum (ER) membrane, thereby causing profound conformational changes that trigger STING oligomerization, its liberation from anchoring factors (such as STIM1), interaction with trafficking factors (for example, SEC24/23 and STEEP), and, finally, incorporation into coatomer protein complex II (COPII) vesicles. While passing through the ER–Golgi intermediate compartment (ERGIC) and Golgi, STING recruits TANK-binding kinase 1 (TBK1), promoting TBK1 autophosphorylation, STING phosphorylation at Ser366, and recruitment of IRF3. The phosphorylation of IRF3 by TBK1 enables IRF3 dimerization and translocation to the nucleus thus inducing expression of type I IFN, IFN-stimulated genes, and several other inflammatory mediators, pro-apoptotic genes, and chemokines.

The activation of STING also leads to NF-κB activation and the formation of LC3^+^ vesicles (autophagosomes) via a non-canonical mechanism of autophagy. In the end, both STING within autophagosomes and from the Golgi apparatus are trafficked to lysosomes, where STING degradation occurs. Steady-state STING translocation through the secretory pathway in the absence of robust cGAMP stimulation is counteracted by continuous retrograde transport to the ER, mediated by COPI vesicles and facilitated by the interaction between STING and SURF4.

Emerging evidence indicates that, in addition to its well-established role in the sensing of cytosolic DNA, the cGAS–STING pathway is involved in restricting RNA virus infection, possibly indicating crosstalk between the innate sensing of cytosolic DNA and innate sensing of cytosolic RNA [211]. Although cGAS is reported to bind dsRNA, this interaction does not lead to cGAMP production [212]. STING interacts with RIG-I and MAVS [213,214], which are key components of the RNA sensing pathway, indicating that it might participate in the RNA virus-induced cytokine production. Although a concerted action of RIG-I and STING in the RNA virus-induced defense responses has been reported [213,215], the underlying mechanism remains unclear.

A recent study suggests that rather than inducing IFN expression, STING initiates global translation inhibition to restrict the production of both viral and host proteins in a RIG-I/MDA5-dependent but MAVS-independent manner [216]. Thus, the recognition of an RNA virus infection by RIG-I/MDA5 probably results in two distinct responses—one is mediated by MAVS to induce IFNs and cytokines, and the other is mediated by STING to inhibit translation.

Growing evidence also indicates that RNA viruses have developed strategies to antagonize cGAS and STING activity. The nonstructural protein NS4B of yellow fever virus (YFV) was the first reported viral protein that inhibits STING through its interaction, although the mechanism is unclear [209]. Additional studies have revealed that hepatitis C virus (HCV) NS4B, a homolog of YFV NS4B, disturbs STING signaling by attenuating the STING–TBK1 interaction [217]. The papain-like proteases from several coronaviruses such as SARS–CoV–1, human coronavirus NL63 (HCoV-NL63), and porcine epidemic diarrhea virus (PEDV) have been shown to associate with STING and block its dimerization and K63-linked ubiquitination, thereby inhibiting the production of IFNβ [218].

To sum up, the utility of STING for mRNA vaccine treatment may lie in its involvement in the signaling initiated by other immune sensors, such as RIG-I. One should probably avoid the activation of cGAS/STING pathway members because it may lower the efficiency of mRNA translation and mediate cytotoxicity. Nonetheless, at present, there are no reports describing the exact involvement of this pathway in the latest mRNA therapies.

Several reports have detailed the structure-based design of cGAS inhibitors because of cGAS’s amenability to crystallography [219,220,221,222,223]. Most of the cGAS antagonists have been discovered by their manner of binding to the active site and thus competing with either ATP/GTP substrates or the cGAMP product. Another major class of cGAS antagonists competes with DNA for coupling with cGAS and interferes with the initial cGAS activation step [224,225,226,227,228].

Currently, there are two main approaches to identifying STING inhibitors. The first one is to design molecules that occupy the cyclic-dinucleotide–binding site and, therefore, serve as competitive antagonists of STING activators. The second approach is to identify inhibitors that bind to either Cys88 or Cys91 near the transmembrane domain of STING, each subject to palmitoylation [229]. In the present literature review, we failed to find reports of experimental assessment of well-known STING inhibitors such as nitrofurans (C-176 and C-178), indole urease (H-151), nitro fatty acids (NO2-FA and NO2-OA), acrylamides (BPK-21 and BPK-25), and brefeldin A and could not determine whether they influence the efficiency of therapeutic mRNA expression. Given that STING acts in a coordinated manner with RIG-I, strategies, addressing the latter may prevent the activation of an immune response involving STING (described above).

cGAS-STING pathway components and their agonists represent promising vaccine adjuvants [230], and their efficacy is currently being tested. Tse et al. have used a constitutively active mutant (V155M) of STING as a genetic adjuvant for encapsulated in lipid nanoparticles (LNP) mRNA vaccines [231]. The mRNA-encoded constitutively active STINGV155M was most effective at maximizing CD8+ T cell responses at the antigen/adjuvant mass ratio of 5:1. The mRNA-encoded STING V155M increased the efficacy of the mRNA vaccines that encode oncoproteins E6 and E7 of human papillomavirus (HPV). The vaccines led to a reduction in HPV + TC-1 tumor growth and thus prolonged the survival of vaccinated mice. The study offers proof of concept of STING application as an mRNA-encoded genetic adjuvant, and this approach requires further evaluation.

cGAS-STING pathway is known to take part in restricting RNA virus infection. Recently, it has also been recognized as a crucial mechanism that regulates tumor immunity [232]. Recently, Miao et al. created STING-activating mRNA-loaded LNPs and elicited vigorous antitumor immune responses in multiple tumor models, including B16-OVA, B16F10, and TC-1 tumors [233]. The ionizable lipids with a cyclic tertiary amine group in the LNPs not only provide adjuvanticity by stimulating the STING pathway but also facilitate cytosolic delivery of mRNA encoding the antigens. Single-dose vaccination with an optimized STING-activating LNPs loaded with OVA-encoded mRNA (A18 mOVA LNPs) cured 3 out of 11 mice of established B16-OVA tumors. STING has proven to be an efficient genetic adjuvant for LNP-encapsulated mRNA vaccines [231].

### 2.7. Endonuclease Activity

It has been widely believed that mRNA decay in higher eukaryotes proceeds mainly at the ends of the molecule along two exonucleolytic pathways [234]. Nowadays, endonuclease activity is proven to be important for mRNA degradation [235]. Endonucleases cut mRNA thus forming short products, which are further degraded by exonucleases.

Exosomes are complexes of ribonucleases, which are present in both the nucleus and the cytoplasm and are involved in mRNA breakdown [236]. Exosomes are thought to destroy mRNA solely due to 3′-5′ exonuclease activity, but it has been demonstrated that one of their components has endoribonuclease activity (e.g., in Rrp44). In addition, SMG6 also engages endoribonuclease activity during mRNA destruction [237]. The discovery of other specific enzymes, such as PMR-1, inositol-requiring enzyme-1 (IRE1), RNase L, aldolase C, activator of RNA decay 1 (ARD-1), Ras GTPase-activating protein SH3 domain-binding protein (G3BP), and apurinic/apyrimidinic endonuclease 1 (APE1), has uncovered their participation in the regulation of individual mRNAs [238].

Nonetheless, this is a hindrance to the development of mRNA vaccines. Their in vivo use requires special modifications to IVT mRNA [239]. For instance, mRNAs should contain recognition sequences such as AU-rich elements (ARE) found in the 3′ untranslated regions of many mRNA transcripts and the coding region determinant of c-Myc mRNA [240,241].

## 3. Applied Aspects: mRNA “Adjuvants” and “Tolerogens”

As mentioned above, mRNA itself can possess some adjuvant properties, when being a part of a vaccine [122,242,243,244]. First and foremost, in vitro synthesized dsRNA and ssRNA can be immunogenic. The former is recognized by TLR-3-related cytoplasmic dsRNA receptors: RIG-I and MDA5 [122]. The latter, if containing unmodified uridine, is recognized by TLR7 [245] and TLR8 [246], and by RIG-I [247] and PKR [248]. Signaling through these receptors causes activation of IRF3, IRF7, and NF-κB, their consequent translocation into the nucleus, and transcription of type I IFN genes (IFN-α and IFN-β) accompanied by the expression of proinflammatory cytokines [249].

There are many ways to reduce the reactogenicity of mRNA in the vaccine. These approaches include efficient purification of IVT mRNA by chromatographic methods [250], the improvement of transcription methods [31,251], polyadenylation of mRNA, and the replacement of uridine with modified nucleotides, which are mostly pseudouridine (Ψ) and m1Ψ [120,123].

Naked RNA is unstable and is quickly degraded by extracellular RNases once it enters the human body [134]. Apart from that, naked RNA cannot be internalized efficiently by the cell. Hence, a variety of transfection reagents have been created to deliver RNA-based therapeutics into the cell, facilitate cellular mRNA uptake, and protect it from degradation [252]. Most RNA transfection agents are characterized by similar adjuvant activity, e.g., protamine (a cationic peptide), O/W cationic nanoemulsion, polyethylene glycol (PEG)-LNPs, a chemically modified dendrimer, polyethyleneimine (PEI), chitosan (polysaccharide), cationic LNPs (in particular containing ionizable cationic lipids), and some combinations thereof [3,253,254]. Taken together, the above observations indicate that mRNA combined with various delivery agents has excellent adjuvant properties.

LNPs used in mRNA-based vaccines activate multiple inflammatory pathways; induce IL-1, IL-1β, and IL-6 [251,255]; and exacerbate inflammation [256]. The FDA has approved LNPs (used in ongoing COVID-19 clinical trials) including a charge-ionizable lipid, a neutral ionizable lipid, a PEG–lipid complex, cholesterol, and DSPC [255]. Parhiz et al. have demonstrated that the major impact of LNPs on inflammation could be attributed to various ionizable lipids. Apparently, they are recognized by TLR4 because LNP-treated TLR4 knockout mice do not mount a significant inflammatory response [256]. This finding is consistent with earlier studies showing the ability of TLR2 and TLR4 to recognize immunostimulatory lipids on the surface of macrophages and other cells [257]. Notably, LNPs with removed ionizable lipids do not initiate inflammation. However, their stability and kinetics of in vivo delivery remain the subject of future research [256]. The immunogenicity of modern LNP-mRNA vaccines can be tailored by changing the lipid composition of LNPs [258,259]. That is probably the reason why the number of studies that addressed the usage of additional adjuvants in RNA-based therapeutics is limited [3,260].

Another promising area of research is the combining of mRNA-LNPs with classic adjuvants to improve the immunogenicity and the persistence of vaccine components at the injection area [260]. The classic adjuvants used in live, attenuated, recombinant, and DNA-based vaccines are listed in Table 2 [260,261,262,263].

The traditional adjuvant MF59 consisting of squalene droplets stabilized with small amounts of surfactants Tween 80 and Span 85 [252] has been employed in different RNA-based therapeutics tested in animal models [272]. Combinations of various classic adjuvants and immune stimulators gave rise to so-called adjuvant systems, such as AS01, AS02, AS03, AS04, and AS15. They are frequently utilized in clinical studies to improve immunogenicity [260,261].

A special place among adjuvants is occupied by genetic adjuvants, which are widely used in DNA-based vaccines (see Table 3) [273,274]. Such adjuvants can be categorized into several groups. The first group comprises cytokines and chemokines. For the purpose in question, the most popular coding sequences are those encoding cytokines IL-2, IL-6, IL-12, IL-15, and GM-CSF, which specifically induce the proliferation of NK cells, B cells, and T cells and the stimulation of dendritic-cell maturation and IFNγ production [275,276,277]. Chemokines IP-10 and CCR7 promote the accumulation and activation of dendritic cells and T cells [278]. The second group contains costimulatory biomolecules, such as CD28, CD40, CD80, and CD86. They specifically induce T-cell proliferation and the maturation of dendritic cells and B cells [279,280]. The third group are genetic adjuvants based on immune-signaling molecules, such as MDA5, RIG-I, MyD88, TRIF, HSP70, NF-κB, IRF1-3-7, T-bet, PD-1, HMGB1, ZBP1/DAI, and TBK1. The last group includes bacterial proteins, namely flagellin [281].

TriMix, another genetic adjuvant, is a mixture of three mRNAs (encoding costimulatory ligand CD70, CD40L, and a constitutively active TLR4) that creates a T-cell–attracting and stimulatory environment, including recruitment of antigen-specific CD4+ and CD8+ T cells. The adjuvant has been used with naked mRNA and mRNA in LNPs and manifested good efficacy [258,282,283,284].

**Table 3 ijms-24-14820-t003:** Examples of genetic adjuvants and their signaling pathways.

Name	Molecular Function	Main Signaling Pathway	Ref.
MyD88	Adaptor protein	MyD88/TRAF6/NFkB	[285,286]
TRIF	Adaptor protein	TRIF/TRAF3/IRF3	[285,287]
IRF1	Transcriptional factor	IRF1 targets	[288,289]
IRF3	Transcriptional factor	IRF3 targets	[288,290]
IRF7	Transcriptional factor	IRF7 targets	[288,290]
HSP70	Chaperone	HSP70-TLR4-NLRP3	[291]
PD-1	Membrane receptor	PD-1/SHP2/STAT1/T-bet	[292]
MDA5	Intracellular receptor	MDA5-TBK1-IRF3	[293]
NF-κB	Transcriptional factor	NF-κB targets	[294]
T-bet	Transcriptional factor	NF-κB targets	[294,295]
RIG-I	Intracellular receptor	RIG-I/TBK1/IRF3	[293]
TBK1	Kinase	TBK1-IRF3	[296]
HMGB1	Signaling protein	HMGB1/TLR4/NF-κB	[297,298]
DAI/ZBP1	Transcriptional factor	DAI/ZBP1 targets	[299]
GM-CSF	Cytokine	GM-CSF/GM-CSFR	[300,301,302,303]
IL-4	Cytokine	IL4/IL4R	[300]
IL-12	Cytokine	IL-12/IL-12R	[304,305,306,307,308,309,310]
IL-2	Cytokine	IL-2/IL-2R	[304,311,312,313]
IL-22	Cytokine	IL-22/IL-22R	[314]
IL-21	Cytokine	IL-21/IL-21R	[315,316]
IL-15	Cytokine	IL-15/IL-15R	[315,316,317,318,319]
IL-6	Cytokine	IL-6/IL-6R	[319]
IL-7	Cytokine	IL-7/IL-7R	[319]
IL-23	Cytokine	IL-23/IL-23R	[320]
CCL3 (MIP-1α)	Chemokine	MIP-1α/CCR1	[321]
CCL20 (MIP-3α)	Chemokine	CCL20/CCR6	[321]
CCL19 (MIP-3β)	Chemokine	CCL19/CCR7	[321,322]
CCL5 (RANTES)	Chemokine	CCL5/CCR5	[319]
CXCL10 (IP-10)	Chemokine	CXCL10/CXCR3	[323]
CCR7	Membrane receptor	CCL19/CCR7	[324,325]
CCL21	Chemokine	CCL19/CCR7	[326]
CD80	Membrane receptor	CD28-CD80	[327,328]
CD86	Membrane receptor	CD28-CD80	[327,328,329]
CD40L	Cytokine	CD40L/CD40	[330,331]
Flagellin	Bacterial protein	TLR5/MyD88/NF-κB	[281]

Genetic adjuvants have been used in several ongoing studies of RNA-based therapeutics. Recent publications describe the use of a fusion of the IgG-Fc domain to the RBD of SARS-CoV-2 as the target antigen within one RNA molecule [258,282,283,284,332,333]. Fc-fusion proteins are commonly used in the clinic and promote immune responses [334]. In another study, researchers used mRNA as a target antigen in combination with mRNA of TriMix adjuvants (CD40L, CD70, and TLR4) within the same LNPs; this approach significantly improved the immunogenic properties of the vaccine [258]. Altogether, these examples represent different approaches to the use of genetic adjuvants in mRNA-LNPs.

Finally, mRNA-LNP vaccines can be utilized for coadministration with antigens and other adjuvants. Different classes of adjuvants (i.e., proteins, peptides, antibodies, antigens, DNA, and small-molecule compounds) can be either encapsulated in LNPs or attached through electrostatic binding or via conjugation with surface molecules [260]. Examples of uses of such combinatorial adjuvants in mRNA-LNP vaccines are limited to a study by Goswami and coauthors [335]. They evaluated mannosylated LNP-mRNA obtained by the incorporation of a stable mannose-cholesterol amine conjugate into the LNPs for ensuring a more rapid onset of an antibody response against an antigen in comparison with unglycosylated LNP-mRNA.

The application of combinatorial adjuvants to LNP-mRNA therapeutics creates a lot of opportunities; however, in vitro and in vivo studies are not sufficient for exploring all the possible options, and supplementary bioinformatic algorithms are recommended for predicting the efficiency of combinatorial adjuvants.

When mRNA is used for non-immunological indications, diminished activation of innate-immunity sensors can improve the expression of the target protein and the efficacy of gene therapy. Therefore, one should strike a balance between high immunogenicity and protein translation efficiency. There are various techniques to achieve this goal (see Table 4).

It should be noted that the described mechanisms of innate immunity in a population can be heterogeneous [336]. Therefore, vaccine reactivity is individual [337]. This fact should be taken into account when formulating new adjuvants/tolerogens that activate different pathways. The option of “balancing” adjuvant and tolerogenic mechanisms should certainly be taken into account in the development of RNA therapeutics including the personalized medicine context.

## 4. Final Remarks

We discussed a multitude of cellular immune responses to RNA invasion. Nonetheless, there are antiviral mechanisms—protein kinase RNA-activated (PKR) and the OAS/RNase L system—which are initiated by the production of interferons. These systems are regarded as one of the most critical aspects in the regulation of gene expression in the presence of dsRNA and should be considered during the development of RNA-based therapeutics.

PKR, also known as eukaryotic initiation factor 2-alpha kinase 2 (eIF2αK2), is a ubiquitously expressed enzyme that is synthesized in response to IFNs [338]. In infected cells, PKR recognizes viral dsRNAs and undergoes dimerization and autophosphorylation. Activated PKR, also called phosphorylated or pPKR, suppresses global translation by phosphorylating eIF2α on Ser51. By phosphorylating other substrates, pPKR also triggers multiple signaling pathways, such as MAPK and NF-κB [339].

Similarly, the presence of interferon triggers the OAS/RNase L system, which is one of the actors in the IFN effector pathways [340]. During viral infection, IFN induces the expression of oligoadenylate synthetase (OAS). It synthesizes 2–5A, which act as second messengers to trigger dimerization and activation of latent RNase L. Activated RNase L cleaves viral and cellular ssRNA and leads to the limitation of virus replication and furthers the apoptosis of infected cells. In addition, many RNA helicases play a critical role during the conformational changes of RNA and ribonucleoprotein (RNP) complexes, and may also have a significant impact on the efficacy of mRNA therapeutics [341,342].

Researchers have already had a wide range of compounds in their arsenal that allow them to manipulate the efficacy of mRNA therapeutics. At academic and industrial research centers, basic research in translation biology and chemistry of nucleic acids makes it possible to create fundamentally new therapeutics and get closer to addressing many socially significant diseases.

Two mRNA vaccines against SARS-CoV-2 have been approved by the FDA recently, but they still pose a risk of side effects, although they are minimal. [343,344]. They may be related to genetic characteristics of immune system reactivity leading to individual adverse reactions to the vaccine [343]. Individual differences in human immunity include polymorphisms in genes encoding TLRs, HLA molecules, cytokines, and cytokine receptors [345]. One of the most troubling adverse effects of SARS-CoV 2 mRNA vaccines that is associated with genetic characteristics is myocarditis [346,347]. The prevalence of this complication is around 0.3–5.0 per 100,000 doses of COVID-19 mRNA vaccines, and it can result in fatality in very rare cases. Recently, the contribution of genetic variants was confirmed in monochorionic diamniotic twins and may be related to HLA alleles [345]. Also, it is important to consider adverse effects like intense homeostatic proliferation and immune imprinting when designing mRNA vaccines [341,348].

Extensive evidence from the Pfizer-BioNTech Vaccine Adverse Effects Analysis suggests that carriers of the HLA-A∗03:01 genotype are likely to experience chills, fever, fatigue, and generally feeling unwell after the vaccination [349]. Taken together, an understanding of individual genetics and the creation of genetic passports can reduce the number of adverse effects of RNA-based therapeutics [350,351].

In conclusion, it can be assumed that the determination of the pattern of immunoreactivity can be adapted for individual selection of adjuvant combinations for the rapid effectiveness of personalized mRNA vaccines for cancer immunotherapy.

## Figures and Tables

**Figure 1 ijms-24-14820-f001:**
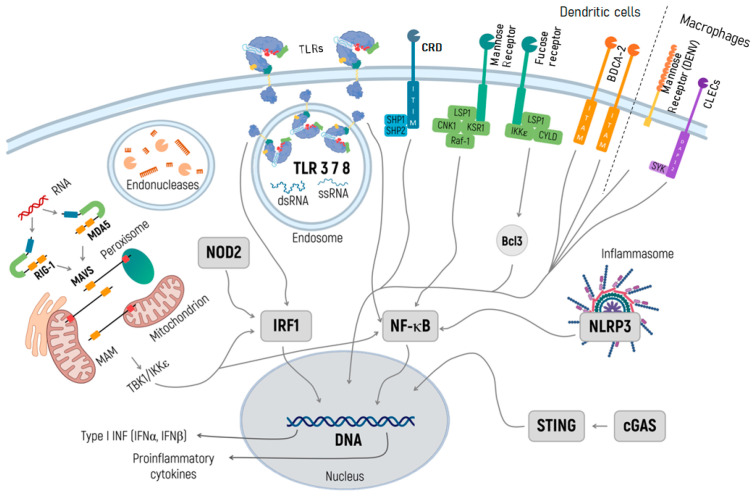
Specific pathways of the innate immune response activation in the context of development and clinical application of mRNA-based therapeutics. TLRs in endosomes recognize both dsRNA and ssRNA and activate transcription factors NF-κB and IRF 1. NF-κB induces inflammation, while IRF3 launches the transcription of IFN genes, which inhibit the translation of exogenous RNA. Through binding to dsRNA and triggering IRF1 and NF-κB, RIG-I and MDA5 launch signaling cascades initiating the transcription of a type I IFN gene. NOD2 recognizes the exogenic RNA and activates IFN gene transcription via IRF1. The inflammasome appears in response to any exogenic particles, particularly delivery systems, and, by activating NF-κB, launches the transcription of proinflammatory cytokines. The cGAS/STING pathway recognizes the DNA and rare RNA and initiates the transcription of proinflammatory cytokines. RIG-I and MDA5 are activated by immunostimulatory RNA, for example, viral RNAs. They then undergo conformational changes that expose and multimerize their CARDs (not shown for simplicity), which allows homotypic CARD–CARD interactions with MAVS. MAVS is anchored with its TM into mitochondria, mitochondrial-associated membranes (MAMs), and peroxisomes, and relays the signal to TBK1 and IκB kinase-ε (IKKε). This signal activates IRF3 and IRF7, which together with the transcription factor NF-κB, induce the expression of type I INF and other genes. When designing RNA-based therapeutics, one should be aware of the endonuclease activity in endosomes, and further in lysosomes, and consider using RNAse inhibitors.

**Figure 2 ijms-24-14820-f002:**
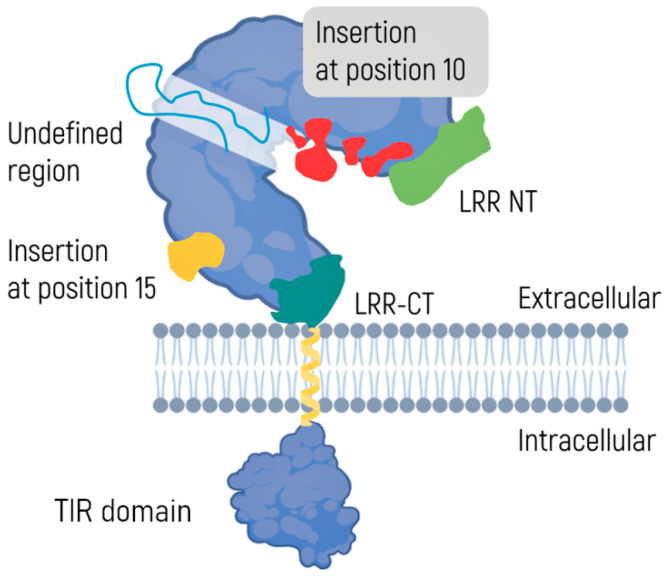
A toll-like receptor: All TLRs are integral membrane glycoproteins with an N-terminal ectodomain and a signal transmembrane domain. The ectodomain of TLR7, TLR8, and TLR9 is depicted, with the LRR solenoid shown with a gray molecular surface, and the N- and C-terminal flanking regions shown in green and purple. Insertions at position 10, indicated in red, might contribute to the information on the PAMP binding site. The insert at position 15, indicated in yellow, is expected to originate on the convex face of the TLR. Also shown is a scheme of the transmembrane domain (presumed to be a single a-helix) followed by a molecular representation of the Toll-IL-1 and IL-18R (TIR) domain [66].

**Figure 3 ijms-24-14820-f003:**
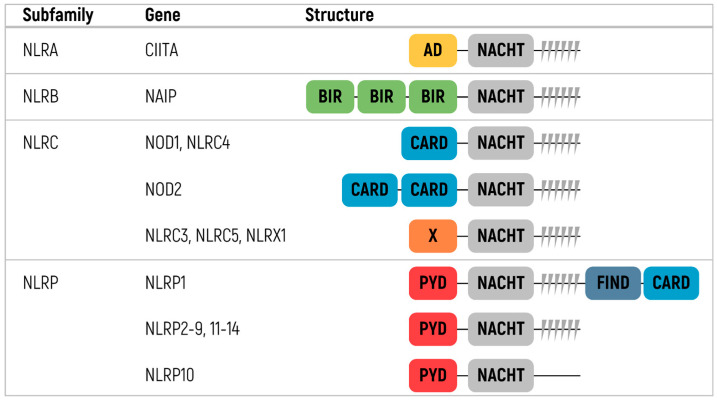
Classification and protein structure of human NOD-like receptor family [8,135]. NLR—NOD-Like Receptors, CIITA—Class II Major Histocompatibility Complex Transactivator, NAIP—NLR family apoptosis inhibitory protein, NOD1—Nucleotide-binding Oligomerization Domain-containing protein 1, NOD2—Nucleotide-binding Oligomerization Domain containing protein 2, NLRP—Nucleotide-binding oligomerization domain Leucine-rich Repeat and Pyrin domain, CARD—Caspase Activation and Recruitment Domain, BIR—Baculovirus Inhibitor of apoptosis protein Repeat, PYD—Pyrin Domain, AD—Activation Domain.

**Figure 4 ijms-24-14820-f004:**
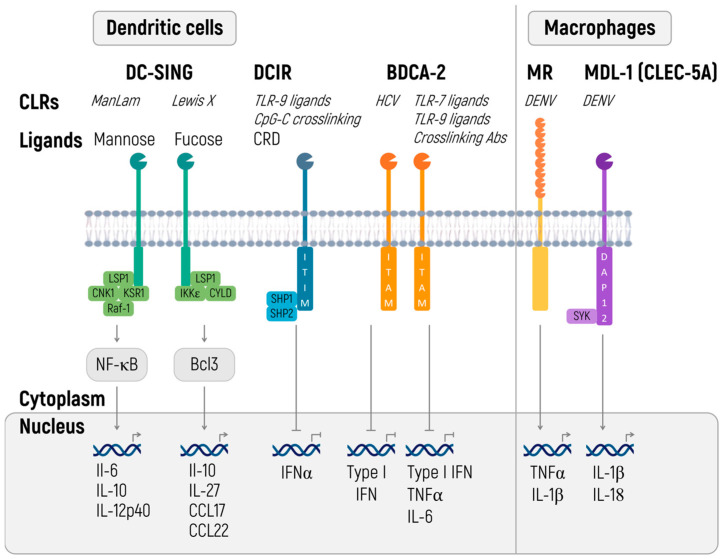
C-type lectin receptors (CLRs) shape innate and adaptive immune responses. CLRs induce innate and adaptive immune responses. Certain CLRs contain ITIM domains and signal via SHP1 and SHP2 phosphatases, whereas other CLRs signal via their ITAM motif. DC-SIGN signaling is carbohydrate specific (CRD) and either signals via Raf-1 signalosome or IKKε and deubiquitinase CYLD, with distinct outcomes [199].

**Table 1 ijms-24-14820-t001:** Biomolecules that are synthesized in response to NF-κB activation.

Biomolecules	Specific Proteins	Roles	Ref.
Cytokines	IL-1, IL-2, IL-5, IL-6, IL-8, Il-12, TNF, G-CSF, GM-CSF, IFN-β, etc.	Mediate the innate immune response	[13]
Chemokines	MCP-1, IL-18, RANTES, MIP-2, CXCL 10, CXCL1	Promote angiogenesis	[14]
Cell cycle regulators	Bcl-2L1, PAI2, cyclins	Stimulate cell proliferation	[15]
Anti-apoptotic factors	Fas, BCL-2, c-FLIP, caspases, IAPs, BFL-1, survivin	Apoptosis regulation	[16,17]
Adhesion molecules	E-selectin, ICAM-1, VCAM-1, ECAM-1, MMPs	Adhesion and cell migration	[18,19]

IL—Interleukin, TNF—tumor necrosis factor, G-CSF—granulocyte colony-stimulating factor, GM-CSF = granulocyte-macrophage colony-stimulating factor, IFN—interferon, MCP = macrophage chemotactic protein-1; MIP = macrophage inflammatory protein-1, RANTES = regulated on activation, normal T-cell expressed and secreted, CXCL—chemokine (C-X-C motif) ligand, Bcl—B-cell lymphoma (Apoptosis regulator Bcl-2), PAI2—Plasminogen activator inhibitor-2, c-FLIP—c-FLICE-like inhibitory protein, IAPs—Inhibitors of apoptosis proteins, BFL-1—Bcl-2-related protein A1, ICAM-1—Inter-cellular adhesion molecule 1, VCAM-1—Vascular cell adhesion molecule 1, ECAM-1—Epithelial cell adhesion molecule-1, MMPs—Matrix metalloproteinases.

**Table 2 ijms-24-14820-t002:** The components of classical vaccine adjuvants.

Adjuvant Category	Component	Ref.
Mineral salts	Alum or calcium salts	[264]
Chemical compounds	Imidazoquinolines, saponin and its derivatives	[265]
Synthetic polynucleotides adjuvants	Poly I:C, CpG oligonucleotides	[261,266]
Adjuvants of bacterial and viral origin	Flagellin, cholera toxin, Bacillus Calmette–Guérin (BCG), virus-like particles, LPS	[267]
Protein-based adjuvants	Proteases, cytokines, chemokines	[268,269]
Emulsions adjuvants	Freund’s adjuvant, incomplete Freund’s adjuvant, and montanides (MF59 and others)	[270,271]

**Table 4 ijms-24-14820-t004:** Examples of methods for reducing the immunogenicity of mRNA, tolerogenic agents, and their signaling pathways.

Name	Main Signaling Pathway	Ref.
Chromatography purification	NLRP3	[94,188]
mRNA capping	RIG-I and MDA5, RLR signaling	[36,47,48,49]
Reduction of mRNA uridine content	RIG-I and MDA5, RLR	[36,47]
Incorporation of modified nucleosides	RIG-I and MDA5, RLR	[36,47,94,119,120,121,122,123,124]
mRNA polyadenylation	RIG-I and MDA5, RLR	[36,47]
Reduction of double-stranded IVT products	RIG-I and MDA5, RLR	[36,47]
Inhibition of PRR	RIG-I and MDA5, RLR	[36,47]
Enhanced DCIR stimulation	TLR	[199]
Increased BDCA-2 stimulation	TLR	[199]
cGAS antagonists	cGAS	[221,222,223,224,225]
Glyburide	NLRP3	[188]
16673-34-0	NLRP3	[188]
FC11A-2	NLRP3	[188]
VX-765	NLRP3	[188]
VX-740	NLRP3	[188]
BHB	NLRP3	[188]
MCC950	NLRP3	[188]
CY-09	NLRP3	[188]
Tranilast	NLRP3	[188]
OLT1177	NLRP3	[188]
Oridonin	NLRP3	[188]
Parthenolide	NLRP3	[188]
Furanochalcone Veluton	NLRP3	[188]

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
