# Peer review of "The Dual Role of the Innate Immune System in the Effectiveness of mRNA Therapeutics"

_ijms, 2023, doi:10.3390/ijms241914820_

Round 1

Reviewer 1 Report (New Reviewer)

Muslimov and colleagues provided a comprehensive review of the innate immune response pathways and the advancements of mRNA as both preventative and therapeutic strategies against infectious diseases, cancers, and other related conditions. While the manuscript is well-written, there are still some issues that need to be addressed.

1. Line 26, The authors should clarify that mRNA is not only used for therapy but also for preventive strategies.

2. Consider restructuring the article by placing sections 2~8 as subsections under a new section titled "Innate Immune Response Pathways recognizing mRNA."

3. In Table 1, some NFκB responsive molecules, such as IL-6, appear to be missing.

4. The receptors in the top right of Figure 1 are not clearly identified, and clarification is needed.

5. Figure 2 seems to repeat information already presented in Figure 1. Consider whether it is necessary to include both figures.

6. Lines 300-303 should mention that TLRs are located on different components of cells for accuracy.

7. Disagree with the statement on line 337 that purified mRNA does not induce IFNs and inflammatory cytokines based on ref. 94. Purified mRNA can still induce cytokine expression, at least at the mRNA level, according to ref. 94.

8. In Figure 4, the authors should provide the full names of each abbreviation in the figure legend.

9. Suggest further dividing section 9 into two different subsections, and consider expanding the content on "tolerogens," as it appears limited.

The quality of English is ok.

Author Response

Thank you for your comments and remarks. We strived to consider all of them.

Below is a list of our point by point replies.

  1. Line 26, The authors should clarify that mRNA is not only used for therapy but also for preventive strategies.

- We changed the phrase in Line 26.

  1. Consider restructuring the article by placing sections 2~8 as subsections under a new section titled "Innate Immune Response Pathways recognizing mRNA."

- The article was restructured according to your suggestion.

  1. In Table 1, some NFκB responsive molecules, such as IL-6, appear to be missing.

- We added several important factors, including IL-6.

  1. The receptors in the top right of Figure 1 are not clearly identified, and clarification is needed.

- We identified all receptors at the top of Figure 1.

  1. Figure 2 seems to repeat information already presented in Figure 1. Consider whether it is necessary to include both figures.

- We deleted Figure 2 because it duplicated data from Figure 1.

  1. Lines 300-303 should mention that TLRs are located on different components of cells for accuracy.

- We mentioned this point in Lines 300-303.

  1. Disagree with the statement on line 337 that purified mRNA does not induce IFNs and inflammatory cytokines based on ref. 94. Purified mRNA can still induce cytokine expression, at least at the mRNA level, according to ref. 94.

- We reworded this statement in light of your point.

  1. In Figure 4, the authors should provide the full names of each abbreviation in the figure legend.

- We provided the full names of every abbreviation in the figure legend.

  1. Suggest further dividing section 9 into two different subsections, and consider expanding the content on "tolerogens," as it appears limited.

- We included all important points in this section of the manuscript, but we didn't divide it to avoid overloading the manuscript.

Reviewer 2 Report (New Reviewer)

Dear authors,

please consider the following comments to improve the content of your manuscript before publication.

The manuscript reviews recent findings on the role of the innate immune system in the development of mRNA therapeutics, with the special emphasis on the fine-tunning of molecular mechanisms in order to achieve the desired effect. Manuscript is well illustrated and documented with up-to-date extensive literature.

However, the last section is termed ‘Discussion’, while it is clear that all previous sections already discussed several important issues. Thus, the section should be renamed into ‘Conclusion’ or ‘Final Remarks’. At the end of the existing text, a paragraph should be added to repeat the major points of the whole manuscript.

Minor Issues

Line 55- change the sentence into “When applying mRNA technology….”

line 56 - “If a eucariotyc cell detects…”

line 155 – Figure 2 legend - please add full term the abbreviation TM

line 172 – instead ‘meaningful’ better term is ‘significant’

line 183-184 – “… dsRNAs activate RIG-I and MDA5 differently: RIG-I and MDA5 preferentially recognize short (not longer than 2 kbp) and long dsRNAs, respectively” please rephrase for clarity

lines 254-255 – figure 3 legend: change term ‘cartoon’ into ‘design’ or ‘scheme’:

line 256 – TIR domain is not labeled on the figure, please add

lines 477-479 - figure 4 does not correspond to the text in these lines. CARD domain is not depicted for NLRP3. Please clarify

line 702 - please add full term the abbreviation LNP, since it is mentioned for the first time

line 778 – please change into “That is probably the reason why…”

Minor editing of English language is needed.

Author Response

Your comments and remarks are greatly appreciated. Thank you very much! We made an effort to consider all of them.

Our point by point replies are listed below.

However, the last section is termed ‘Discussion’, while it is clear that all previous sections already discussed several important issues. Thus, the section should be renamed into ‘Conclusion’ or ‘Final Remarks’. At the end of the existing text, a paragraph should be added to repeat the major points of the whole manuscript.

  • We renamed this section into ‘Final Remarks’

Minor Issues

Line 55- change the sentence into “When applying mRNA technology….”

changed

line 56 - “If a eucariotyc cell detects…”

changed

line 155 – Figure 2 legend - please add full term the abbreviation TM

  • We added full terms for Figure legend;

line 172 – instead ‘meaningful’ better term is ‘significant’

changed;

line 183-184 – “… dsRNAs activate RIG-I and MDA5 differently: RIG-I and MDA5 preferentially recognize short (not longer than 2 kbp) and long dsRNAs, respectively” please rephrase for clarity

- We rephrase this sentence;

lines 254-255 – figure 3 legend: change term ‘cartoon’ into ‘design’ or ‘scheme’:

changed;

line 256 – TIR domain is not labeled on the figure, please add

  • We changed the Figure;

lines 477-479 - figure 4 does not correspond to the text in these lines. CARD domain is not depicted for NLRP3. Please clarify

  • We corrected the sentence;

line 702 - please add full term the abbreviation LNP, since it is mentioned for the first time

  • We provide full name of the abbreviation;

line 778 – please change into “That is probably the reason why…”

changed;

Reviewer 3 Report (New Reviewer)

The review by Muslimov et al., addresses an important question however it needs significant improvement before being accepted.

1. At many places there is use of non-scientific language (one example line 47, fabricate). 

2. Line 48; authors should elaborate the merits and demerits in the same section.

3. References are missing (line 75, line 86 and several other places)

4. Figure 1 is extremely simplified and lacks details. The authors should include some if not all receptors and more details like mechanism of NLRP3 activation.

5. Line 170, the authors should explain type of modification done at mRNA level as that is an integral part of mRNA vaccine platform. Also key references are missing.

6. Line 855; Authors need to be very careful when discussing about myocarditis and other complications and should include a separate section elaborating and explaining what are causal factors for fatal reactions. They should also include statistics as the percentage of people having adverse effect and the underlying cause. Generalizing adverse reactions can lead to spread of misinformation.

Language editing is needed

Author Response

Thank you for your comments and notices very much. We made an effort to take into account all of them.

  1. At many places there is use of non-scientific language (one example line 47, fabricate). 

- We changed several terms, including ‘fabricate’;

  1. Line 48; authors should elaborate the merits and demerits in the same section.

- We would like to keep this paragraph without merits and demerits description as these points meticulously discussed below;

  1. References are missing (line 75, line 86 and several other places)

- The necessary references were added;

  1. Figure 1 is extremely simplified and lacks details. The authors should include some if not all receptors and more details like mechanism of NLRP3 activation.

- We changed the figure by adding some information;

  1. Line 170, the authors should explain type of modification done at mRNA level as that is an integral part of mRNA vaccine platform. Also key references are missing.

-We added an explanation;

  1. Line 855; Authors need to be very careful when discussing about myocarditis and other complications and should include a separate section elaborating and explaining what are causal factors for fatal reactions. They should also include statistics as the percentage of people having adverse effect and the underlying cause. Generalizing adverse reactions can lead to spread of misinformation.

- We rephrased these statements and provided additional information;

Reviewer 4 Report (New Reviewer)

The manuscript titled “The dual role of the innate immune system in the effectiveness of mRNA therapeutics” is a detailed and extensive review of the immune responses and pathways involved in mRNA therapeutics.  Each section describes different signaling pathways in detail and the authors have done a thorough job in providing details and discussion in each section.  The manuscript is recommended for publication with minor edits.

The authors have to include a future perspectives discussion part to provide insights into future of mRNA therapeutics in terms of addressing immune responses to immunotherapy and non-immunogenic modalities. 

What are some ways to address protective immunity vs reactogenicity? Please provide examples of therapeutics in preclinical or clinical phase, whenever applicable in this discussion.

Author Response

Thank you for your comments and remarks and evaluation of our manuscript.

The authors have to include a future perspectives discussion part to provide insights into future of mRNA therapeutics in terms of addressing immune responses to immunotherapy and non-immunogenic modalities. 

What are some ways to address protective immunity vs reactogenicity? Please provide examples of therapeutics in preclinical or clinical phase, whenever applicable in this discussion.

-         Thank you for these suggestions. However we would like to keep the manuscript without separate section with perspectives discussion and repeate points of ‘reactogenic’ and ‘tolerogenic’ ways for vaccine design as these issues discussed in the body of manuscript in detail.

Round 2

Reviewer 3 Report (New Reviewer)

The authors have addressed my concern.

Minor editing of language needed

This manuscript is a resubmission of an earlier submission. The following is a list of the peer review reports and author responses from that submission.

Round 1

Reviewer 1 Report

This is a review of the associated mechanisms of the innate immune system

in the effectiveness of mRNA drugs. It could be an interesting review if the authors managed to write and organize the manuscript to make it more readable, right now it is hard to read and understand without supporting figures.

Some aspects could be improved:

1. I don't think the term “drugs” is appropriate to designate medicine for therapeutic use.

2. The authors describe their work in the abstract as a mini-review, however the length is a review.

3. In the introduction, it is necessary to use bibliographical references to argue the claims that the authors make.

4. The introduction could be focused and reduced, it is hard to read.

5. The organization of the manuscript in paragraphs makes it easier to read.

6. The figures should be self-explanatory, with a caption indicating the abbreviations and the different steps to follow. Fig1. An unlabelled blue square appears.

7. In the different sections of the review, it would be advisable to include tables and figures that allow breaking down and summarizing the text included by the authors.

8. line 172. Thectivation?

9. Table 1. Genetic adjuvants??? The authors show proteins that perform a molecular function.

10. Table 2. Genetic tolerogens??? The authors show different laboratory techniques and chemical components... the chromatographic purification technique should be detailed. Acronyms should be included at the bottom of the table.

11. “N of 1” should be explained.

12. The limitations of the use of mRNA should also be discussed by the authors in the introduction.

Reviewer 2 Report

Muslimov et al. present a review article focussing on the many different aspects of the intracellular innate immune system, specifically those that trigger inflammatory and IFN patterns in the context of mRNA-based vaccines.

Overall the manuscript addresses most of the aspects involved in the PAMP/DAMP-PRR induction. Authors are however missing to comment and discuss the possible role of PKR and OAS/RNase L role. The authors also fail to address the problem of triggering the pathways they explain in a cell-specific context, which may change the outcome of mRNA vaccines.

The article requires extensive editing, including the consideration of being able to write concise, short paragraphs that address the issue they want to raise and not explain the entire specific signaling pathway. There are a lot of examples of slopiness, including the absence of referring to dsRNA when talking about RNA, lack of the correct verb sense, presence of Cyrillic characters, etc. Extensive editing is strongly suggested. The list of grammatical issues is long.

Many parts of the text are well referenced, containing more information than the one that addresses the issue of the review (ie: authors mention the many possible NF-kB heterodimers while they do not place clear concepts on the specifics of mRNA vaccination. In some other parts of the text (ie: the STING section, the role of MAPK/AP-1 in IFN and proinflammatory cytokine induction...), the authors lack many references.

The authors mention RIG-I and MDA5 as RLR, but they don´t comment on ent the regulatory functions of LGP2.

Authors mention PKR and RNase L (OAS/RNase L?) in the text, being one of the most critical aspects in the regulation of gene translation in the presence of dsRNA. In addition, protein synthesis shutoff is not only triggered by dsRNA, also by ER stress as well as other mechanisms that can be derived from mRNA vaccination. The authors don´t address either the possible role of the many RNA helicases in the biology of vaccine mRNA vaccines.

Authors should consider the use of italics or underlined names when referring to pathogens (i.e.: Mycobacterium tuberculosis, E. coli.), as well as the use of Latinisms like "in vivo"...

NOD-like receptors are intracellular PRR (line 340), not sure whether they are referring to TLR in this first part of the paragraph. Also, the authors talk about NLRs in section 4 and come back talking about NLRP3 in section 7. Section 4 is poorly described and requires more attention.

While some parts of the text are very long and lack focus, the discussion is scarce, commenting on the 3 aspects: composition of the vaccine, human polymorphisms/pathology, and consideration of the industrial scale and impact of these vaccines. Many different aspects are inferred or addressed along text that should be moved or commented on in the discussion.